# Ambulatory Blood Pressure Monitoring for Diagnosis and Management of Hypertension in Pregnant Women

**DOI:** 10.3390/diagnostics13081457

**Published:** 2023-04-18

**Authors:** Walter G. Espeche, Martin R. Salazar

**Affiliations:** 1Unidad de Enfermedades Cardiometabólicas, Hospital General San Martín, La Plata 1900, Argentina; 2Docencia e Investigación, Hospital San Martin de La Plata, La Plata 1900, Argentina

**Keywords:** preeclmpaisa, hypertension disorder pregnancy, ambulatory blood pressure monitoring, masked hypertension, nocturnal hypertension

## Abstract

Hypertension disorders during pregnancy has a wide range of severities, from a mild clinical condition to a life-threatening one. Currently, office BP is still the main method for the diagnosis of hypertension during pregnancy. Despite of the limitation these measurements, in clinical practice office BP of 140/90 mmHg cut point is used to simplify diagnosis and treatment decisions. The out-of-office BP evaluations are it comes to discarding white-coat hypertension with little utility in practice to rule out masked hypertension and nocturnal hypertension. In this revision, we analyzed the current evidence of the role of ABPM in diagnosing and managing pregnant women. ABPM has a defined role in the evaluation of BP levels in pregnant women, being appropriate performing an ABPM to classification of HDP before 20 weeks of gestation and second ABMP performed between 20–30 weeks of gestation to detected of women with a high risk of development of PE. Furthermore, we propose to, discarding white-coat hypertension and detecting masked chronic hypertension in pregnant women with office BP > 125/75 mmHg. Finally, in women who had PE, a third ABPM in the post-partum period could identify those with higher long-term cardiovascular risk related with masked hypertension.

## 1. Introduction

Hypertension disorders during pregnancy (HDP) are the leading causes of neonatal and maternal morbidity and mortality worldwide [1,2,3,4]. However, HDP has a wide range of severities, from a mild clinical condition to a life-threatening one. Pre-eclampsia (PE) and eclampsia represent the most severe forms of HDP [5].

The normal physiological decrease in systemic vascular resistance leads to a decrease in BP, with its nadir at 16–18 weeks of gestation, followed by a return to prepregnancy levels by the third trimester. The decrease in diastolic blood pressure (by as much as 20 mm Hg) is more marked than the decrease in systolic blood pressure. [6] (Figure 1). Despite these physiological changes in BP levels, the threshold recommended to define hypertension in pregnant women (regardless of the weeks of gestation) is the same as that used to define hypertension in the general population [7,8,9,10,11]. However, given that the 140/90 mmHg cut point was defined based on studies developed in the general population, pregnant women are clearly underrepresented. Furthermore, this cut point was estimated for the prevention of macrovascular events (such as acute myocardial infarction or stroke) but not for the prevention of specific pregnant complications such as PE, eclampsia, and HELLP syndrome. Additionally, in the last decades, new technologies, such as ambulatory blood pressure monitoring (ABPM) and home blood pressure monitoring (HBPM), have allowed the evaluation of out-of-office BP. Thus, new hypertension phenotypes, such as masked and withe coat hypertension, were added into medical practice, and their importance was widely shown in the general population. However, there are few studies regarding the prognostic of these phenotypes in pregnancy. 

Despite this limitation, in clinical practice office BP of 140/90 mmHg cut point is used to simplify diagnosis and treatment decisions. Furthermore, out-of-office BP evaluations are not widely used and limited when it comes to discarding white-coat hypertension. In this revision, we analyzed the current evidence of the role of ABPM in diagnosing and managing pregnant women.

## 2. Out-of-Office Blood Pressure in Hypertension Diagnosis in Pregnant Women

### 2.1. White-Coat Hypertension

Currently, office BP is still the main method for the diagnosis of hypertension during pregnancy [11,12,13]. However, when office BP is ≥140/90 mmHg, out-of-office measurements should be performed to discard white-coat hypertension. White-coat hypertension is a condition in which a patient has high blood pressure in the office but normal BP outside the office. In clinical practice, 24-h ABPM or HBPM may be used to identify white-coat hypertension. The threshold recommended to define hypertension in pregnant women (regardless of the weeks of gestation) is the same as that used to define hypertension in the general population [14]. The reported prevalence of white-coat hypertension in pregnancy has been inconsistent in the literature, ranging from as low as 4% [15,16] to 30% [17,18]. White-coat hypertension in pregnancy may have a relatively good prognosis [18]. Due to the benign nature of white-coat hypertension, compared with sustained hypertensions, the diagnostic confirmation with ABPM or HBPM is useful to avoid unnecessary intensified treatment that could be detrimental to utero-placental perfusion. Moreover, this misinterpretation may also occur in treated pregnant women (white-coat effect). Thus, an elevated office BP could be interpreted as a lack of response to treatment and/or a hypertensive emergency, leading to overtreatment with hypotensive drugs and/or an unnecessary emergency interruption of pregnancy [17,18]. In consequence, AMBP monitoring or HBPM should be performed on any pregnant women with office BP > 140/90 mmHg, regardless of the weeks of gestation.

### 2.2. Masked Hypertension

Until recently, out-of-office BP measurements were not recommended in pregnant normotensive women according to office BP values. However, masked hypertension is a common condition in the general population [19], and the behavior of BP described above during the first half of pregnancy suggests that it could be even more frequent in pregnant women. The prevalence of masked hypertension is unknown in normal pregnancies. Our working groups have shown that ~30% of high-risk pregnancies have masked hypertension [20]. Moreover, masked hypertension is a strong and independent predictor for the development of PE and poor neonatal outcomes [20,21,22]. Thus, in high-risk pregnant women, the detection of masked hypertension is necessary.

Although masked hypertension is mentioned in the current clinical practice guidelines, no specific recommendations regarding the use of ABPM in pregnant women with normal BP values in the clinical evaluation are provided [7,8,9,10,11]. Although discarding white-coat effect is relatively simple (performing an ABPM in all women with office BP ≥ 140/90 mmHg), searching for masked hypertension seems a more complicated task. Masked hypertensive patients have by definition, a ‘normal’ office BP, therefore, some questions arise: Is 140/90 mmHg an appropriate hypertension threshold for pregnant women? Moreover, what would be the office BP values in pregnant women below which out-of-office BP measurements would not be needed? Regarding these issues, Wu et al. [23], using a low-risk cohort of 47,874 cases, reported that women with 130–139 mmHg systolic BP and/or 80–89 mmHg diastolic BP diagnosed in early gestation had significantly increased incidences PE. In addition, Salazar et al. [16] have shown that in high-risk pregnant women with office BP values above 125–75, the risk of developing PE increases significantly (Table 1).

In summary, current evidence suggests it is necessary to use out-of-office BP measurements more widely, including women with office <140/90 mmHg, in order to detect masked hypertension. Recently Salazar et al published revision suggested that in women with high-risk pregnancies office BP ≥ 125/75 mmHg and ABPM should be performed [16]. However, if this approach is valid to normal a pregnancy remains to be established.

Ambulatory blood pressure monitoring vs. home blood pressure monitoring.

Two methods are currently used to evaluate the levels of out-of-office BP, AMPM and HBPM, which have some differences [24]. While ABPM evaluates BP during 24 h, including day activities and nocturnal rest, HBPM evaluates BP during a more prolongated period, usually a week, and (except for more recent developments) does not include a nocturnal period [25,26]. In non-pregnant women, ABPM is more suitable for diagnosis and HBPM is used to improve hypertension control. No head-to-head comparisons of both methods are available in pregnant women, and most current evidence is provided from studies using ABPM.

We think that ABPM has some advantages over HBPM in the evaluation of pregnant women. First, ABPM allows the evaluation of BP levels during nocturnal rest. In women with high-risk pregnancy, nocturnal hypertension is the most common abnormality of all hypertensive disorders; being indeed essentially a masked condition. Remarkably, ~20% of pregnant women with normal values in 24-h ABPM have nocturnal hypertension, or isolated nocturnal hypertension [20]. Nocturnal hypertension is a strong predictor of PE (five times greater risk) [21], which may constitute an early finding, several weeks before clinically evident disease [20,21]. In consequence, to detect abnormal BP during nocturnal rest seems important in high-risk pregnancy. In second place, HBPM implies BP measurements during one week and could be inappropriate to use for making timely clinical decisions [27]. Therefore, we prefer ABPM over HBPM to evaluate out-of-office BP in the diagnosis of hypertension in pregnant women.

## 3. Timeliness of Performing Out-of-Office Blood Pressure Measurements

### 3.1. ABPM before 20 Weeks of Gestation: Identifying Chronic Hypertension

Traditionally, hypertensive disorders of pregnancy have been divided, using the office blood pressure (BP), into 1-chronic arterial hypertension (women who had hypertension that become pregnant) and 2- gestational hypertension (pregnancy-induced hypertension) [7,8,9,10,11]. Thus, according to the traditional definitions a pregnant woman without antecedents of hypertension with an office BP < 140/90 mmHg before 20 weeks of gestation who subsequently develops hypertension should be defined as having gestational hypertension (Figure 2). On the other hand, a woman without antecedents of hypertension who has office BP ≥ 140/90 mmHg before 20 weeks of gestation should be considered as having chronic hypertension. However, this approach has some debilities.

The first consideration is the fact that not much is known about hypertension by the general population around the world. Only one in three individuals who have hypertension are aware of their diagnosis, and this value is even lower in young people [28]. Thus, the possibility that young women with hypertension ignore their condition is very high. In consequence, all women should have a BP evaluation ideally before conception (due to the physiological decrease of BP described above), or at least in the first half of pregnancy. The second question is about the method used to evaluate the BP. Is it sufficient to evaluate using only office readings? Office BP do not identify masked hypertension; consequently, these women could be erroneously classified as normotensive. In this regard, Espeche et al. [29] recently published a study which found that ~60% of women who develop gestational hypertension (diagnosed according to traditional approach, office BP < 140/90 mmHg) had indeed masked chronic hypertension if the values of an ABPM performed before 20 weeks of gestation are included in the evaluation (24 h ABMP ≥ 130/80 mmHg with normal office BP). Moreover, these women had a very high-risk for developing PE.

The clinical significance of different hypertensive disorders of pregnancy is not the same. While the risk of maternal and fetal complications of chronic hypertension has been shown [30], the risk associated with gestational hypertension is less well defined. A previous published meta-analysis of 92 cohort studies found that chronic hypertension was the second most significant risk factor for the development of PE after PE in previous gestation [31]. Wu et al. [32], using the National Inpatient Sample database, analyzed the association between different hypertensive disorders of pregnancy and adverse in-hospital maternal and fetal outcomes in more than 44 million deliveries. Women with chronic hypertension, but not those with gestational hypertension, had a higher risk of both maternal and fetal adverse outcomes [30]. The International Society for the Study of Hypertension in Pregnancy (ISSHP) in their last position paper state that “outcomes in pregnancies complicated by gestational hypertension are normally good, but about a quarter of women with gestational hypertension will progress to preeclampsia and have poorer outcomes”. They also stated that “gestational hypertension is not a uniformly benign condition” [13]. The risk of complications has been attributed to the gestational age at which it develops [33]. In addition, some data suggests that different outcomes could be related to different out-of-office BP levels. Davis et al. found that pregnant women with gestational hypertension who developed preeclampsia/eclampsia had higher ABPM values than those who did not [34]. These data suggest that some of this heterogeneity in the prognosis of gestational hypertension could be due to the fact that some women misclassified as gestational hypertension had indeed masked chronic hypertension.

Thus, the appropriate identification of pregnant women with chronic hypertension, including those with masked chronic hypertension, is clinically significant, and to perform an ABPM in the first half of pregnancy seems an appropriate approach to accomplish this task.

### 3.2. ABPM after 20 Weeks of Gestation: Identifying Risk of Preeclampsia/Eclampsia

The development of PE is the leading complication of the HDP producing a significant burden of maternal and neonatal morbidity and mortality [35]. PE (and the convulsive phase, eclampsia) occurs during the second half of pregnancy and more frequently in women chronic hypertension (unmasked or masked) than those with gestational hypertension. Identifying women’s risk of developing PE is a very important task and different approaches have been proposed. Several clinical factors have been associated with the development of PE [36]. However, none of these are enough to identify all these women.

An imbalance between angiogenic and antiangiogenic factors have been proposed as causes for the development of PE. In normal pregnancy, appropriate extravillous trophoblast invasion into the maternal endometrium leads to sufficient maternal blood flow from the spiral artery. The placental growth factor (PlGF), which is secreted from the placenta, activates the vascular endothelial growth factor (VEGF) and maintains a healthy endothelium. On the other hand, PE begins with abnormal trophoblast invasion and spiral artery remodeling before clinical manifestations of the disease become apparent. In preeclamptic pregnancy, incomplete invasion of the extravillous trophoblast leads to insufficient maternal blood flow from the spiral artery and subsequent placental hypoxia. Consequently, soluble fms-like tyrosine kinase-1 (sFlt1) is secreted from the placenta, which suppresses VEGF, resulting in systemic endothelial in multiple organs dysfunction, which are the cause of main clinical manifestations of preeclampsia such as hypertension, proteinuria, seizures, and liver dysfunction [37,38].

Serum markers (prostaglandins, cytokines, VEGF soluble receptor-1 sFlt-1-.) of this imbalance have been proposed for identifying women at risk for PE; the ratio of sFlt-1/PlGF appears as particularly promising in high-risk pregnancy [38]. The sFlt-1/PlGF ratio or PlGF alone can be used to exclude preeclampsia within 7–14 days because both have good negative predictive value for up to 4 weeks.

Traditionally, studies of uterine artery Doppler velocimetry were proposed for prediction of early preeclampsia in the first trimester of gestation [39,40]. However, studies with Doppler are difficult to standardize because investigators have used different Doppler sampling techniques, definitions of abnormal flow velocity waveform, gestational age at examination, and criteria for the diagnosis of preeclampsia. Moreover, uterine artery Doppler findings should not be interpreted alone, but rather in combination with other clinical/demographic risk factors, or serum biomarkers.

Since no single clinical test is powerful enough to identify all women who will develop PE, the Fetal Medicine Foundation (FMF) [41] have proposed the combination of PIGF with clinical maternal factors, measurements of mean arterial pressure, and the uterine artery pulsatility index. A recently published study showed that the first-trimester prediction model (the FMF triple test) had high detection rates for the prediction of early and preterm preeclampsia. However, this study was performed on Asian women and the reproducibility of these results in different ethnic groups have not been shown. Moreover, PIGF determinations are not widely available in low-middle income countries.

Blood pressures values are a traditional and widely accessible markers of the development of PE. Office hypertension is related to maternal and fetal mobility and mortality. However, a substantial proportion of the risk for developing preeclampsia/eclampsia remains in pregnant women with office BP values less than 140/90 mmHg [42]. In this context, ABPM monitoring has emerged as a promising tool for identifying the risk of PE.

Our working groups have demonstrated the usefulness of ABPM to predict PE. In a study including high-risk pregnant women using ABPM, performed in ~30 weeks of gestation, we showed that adjusted relative risks increased with the presence of nocturnal (odds ratio = 4.72, 95% confidence interval 1.25–19.43, *p* = 0.023) or masked hypertension (odds ratio = 7.81, 95% confidence interval 2.6–22.86, *p* = 0.001). Remarkably, nocturnal systolic BP and diastolic BP had the highest abilities to predict PEEC (area under the curve = 0.77 and 0.80, respectively) [20]. Subsequently, we showed that an ABPM at the mid-pregnancy (23 ± 2 weeks of pregnancy) has a similar predictive value. Interestingly, this risk doubles if the women did not consume a preventive low dose of aspirin (OR 11.40 95% CI 2.35–55.25) [21]. This modulating effect of aspirin use suggests that nocturnal hypertension could be an indication for aspirin treatment. Moreover, masked hypertension, but not masked white-coat hypertension, increases the risk for poor neonatal outcomes (adjusted OR 2.58 95% CI 1.23–5.40) [22].

In an Editorial of the *Journal of Hypertension*, Bilo and Parati [43] stated that masked and nocturnal hypertension, despite normal office BP, is a strong predictor of PE development in high-risk women. The authors highlights the importance of the routine use of ABPM to identify women likely to progress to PE. Thus, the routine use of ABPM at 20–30th weeks of gestation can identify women likely to progress to PE. Also, based on these findings, ABPM should be preferred top HBPM because, at least with the commonly available device technology, HBPM is unable to provide information on nocturnal BP.

## 4. Nocturnal Hypertension and Risk for Early Onset Preeclampsia

### 4.1. Early vs. Late Onset Preeclampsia

In 1996, Ness and Roberts [44] published a seminal paper highlighting the heterogeneity of preeclampsia, suggesting that “maternal preeclampsia” differs from “placental preeclampsia” based on the extent of trophoblastic invasion during the first and early second trimesters of pregnancy, as well as rates of fetal growth restriction. They suggested that “placental preeclampsia” was linked to early-onset PE but that “maternal preeclampsia” was linked to late-onset PE [43]. Thus, in early onset PE placental abnormalities may cause chronic uteroplacental insufficiency, local ischemia, and the release of inflammatory cytokines, resulting in earlier maternal hypertension in early-onset preeclampsia. On the other hand, late-onset preeclampsia is more frequently based on placental dysfunction associated with chronic oxidative stress due to maternal metabolic abnormalities, such as obesity and insulin resistance [45]. There is an agreement that a cut-off point of 34 weeks of gestation differentiates between early and late onset PE [46,47,48].

Published data supports the idea that early and late onset PE are different phenotypes of pregnancy with different maternal and fetal consequences. In a population-based study of ~450,000 pregnancies in Washington State between 2003 and 2008, Lisonkova et al. [49] showed that risk factors differ significantly in their association with early vs. late- onset PE, with the background for chronic hypertension having a stronger association with early-onset PE than with late-onset PE. Remarkably, this study showed that the rates of adverse birth outcomes were significantly higher among women with early onset PE. Furthermore, very low birth weight and perinatal death were several times more frequent in early-onset than in late-onset PE. Thus, identifying women who may develop early onset PE has great clinical importance, as most of the burden of maternal and fetal disease occurs with this phenotype.

### 4.2. Nocturnal Hypertension and Risk for Early Onset Preeclampsia

A recent published paper showed that nocturnal hypertension was the most prevalent finding and that it was highly prevalent in women who developed early onset PE (88.6%) and only 1.6% of women without nocturnal hypertension developed early onset PE [50]. Additionally, nocturnal hypertension was a stronger predictor for early onset PE than for late-onset PE (adjusted OR, 5.26 95% CI 1.67–16.60) vs. 2.06, 95% CI 1.26–4.55, respectively). Interestingly, the detection of nocturnal hypertension preceded the development of early PE in about 4 weeks [50].

Thus, nocturnal hypertension is not only a prevalent BP abnormality, but it is also a strong predictor of early onset PE in high-risk pregnancy. Moreover, nocturnal hypertension can herald the development of PE by several weeks.

High nighttime BP is associated with a risk of cardiovascular events in both the general and hypertensive populations. In the Japan Morning Surge-Home Blood Pressure (JHOP) study, both N-terminal-proBNP (NT-proBNP) and nighttime BP were associated with cardiovascular events, and nighttime home BP could mediate the association between elevated NT-proBNP and cardiovascular events [51]. The increased nighttime BP could produce cardiac and renal damage, which may lead to volume overload. Also, the production sFlt-1 for a pathological placenta may increase BNP, resulting in nocturnal hypertension [52].

## 5. Hypertension Control in Pregnant Women with Chronic Hypertension

The management of HDP under pharmacological treatment is a matter of debate. Although the development of maternal and fetal complications is strictly related to office BP values (the greater the severity of hypertension, the greater the possibility of developing maternal or fetal events) [7,8,9,10,11], the therapeutic goals have not been sufficiently studied. It has been shown that BP values > 160–100 mmHg altered the cerebral autoregulation and cerebral edema, which begins to be a frequent finding [53,54]. Likewise, it has recently been shown that severe hypertension has the same maternal-fetal morbidity and mortality as PE [55]. Therefore, the therapeutic indication is indisputable in HDP when the BP office exceeds 160–100 mmHg. However, in pregnant women with mild hypertension (<160–100 mmHg) the controversy regarding drug treatment is a matter of debate. In some studies performed on pregnant women with mild (not severe) hypertension, pharmacological treatment does not improve maternal and fetal hard events (maternal and fetal death, placental abruption, or acute pulmonary edema) [56]. Moreover, there are some concerns about that antihypertensive treatment during pregnancy may restrict the fetal growth and increase the risk of preterm delivery [57].

However, recently published data suggests that untreated hypertension, not antihypertensive medication, is the true risk for the mother and child. Tita et al. [58], in an open-label, multicenter, and randomisedstudy, showed that the pharmacological treatment of pregnant women with mild chronic hypertension (BP goal < 140/90 mm Hg) reduces the risk of PE (risk ratio, 0.79; 95% CI, 0.69 to 0.89), with no increase in risk for small-for-gestational-age birth weight. Also, a recent metanalysis showed that blood pressure-lowering treatment significantly prevented not only severe hypertension, preeclampsia, and severe preeclampsia, but also placental abruption and preterm birth, although the risk of small for gestational age was increased [59]. The KAMOGAWA study group, a study multicentric study performed in Asian women with chronic hypertension, showed that a systolic BP < 130 mmHg within 14–15 weeks of gestation reduces the risk of early onset superimposed preeclampsia [60].

Consequently, most current data supports the possibility that an appropriate treatment and control of women with chronic hypertension who become pregnant could prevent the development of PE, particularly early onset PE. Up to now, the role of out-of-office measurement and ABPM in the evaluation of BP control in pregnant women was not defined. However, some consideration should be made. The correlation of office BP measurements with ABPM is poor in pregnant women in the last trimester of pregnancy [61]. Likewise, as we have shown above in this revision, the prevalence of white-coat hypertension and masked hypertension is high-risk in pregnant women. This could also be observed in women under treatment (namely, white-coat effect and masked uncontrolled hypertension). Preliminary unpublished data of our cohort about women with high-risk pregnancies suggest that ABPM, and especially nighttime BP levels, are better than office BP to evaluate hypertension control and risk for PE in pregnant women with treated chronic hypertension. The relative risk for PE increased 5% and 6% for each mmHg of daytime systolic and diastolic ABPM, and 6% and 7% for nighttime systolic and diastolic ABPM, respectively. However, when the risk was adjusted for the levels of ABPM in the opposite period of the day (daytime BP by nocturnal BP and nocturnal BP by daytime BP), only nocturnal systolic and diastolic ABPM remains significant as predictors of PE. Moreover, office BP levels were not independent predictors.

## 6. Post-Partum Ambulatory Blood Pressure and Long-Term Cardiovascular Risk in Pregnant Women

Traditionally, cardiovascular disease was considered more prevalent in men than in women [61]. In the last decade, worldwide, the absolute number of deaths from cardiovascular events has declined in men parallel to the improvement in the control of the traditional risk factors. However, this improvement has not been of the same magnitude in women [62]. A possible (at least partial) explanation for this difference between the sexes could be the effect of non-traditional risk factors. It has been shown recently that HDP, especially PE, increases the long-term risk of cardiovascular disease, possibly through occurrence of hypertension after delivery including masked hypertension and nocturnal hypertension. Both white-coat and masked hypertension need out-of-office BP measurements to be properly diagnosed. Benschopet al [63] recently showed that, one year after delivery, 41.5% of pregnant woman who had had PE still showed signs of hypertension in an evaluation using ABPM. The most common phenotypes of hypertension was masked hypertension (17.5%), followed by sustained hypertension (14.5%), and white-coat hypertension (9.5%). Therefore, in pregnant women who develop HDP, especially PE, a post-partum evaluation with ABPM could be useful in order to detect masked hypertension and to prevent future cardiovascular disease.

## 7. Future Scope and Limitations

Some limitations in our knowledge about the utility of ABPM in pregnant women that must be addressed.

Most evidence comes from observational studies and has inherent limitations, a prospective randomized clinical trial would be necessary to confirm the findings.Most evidence comes from studies performed on women with high-risk pregnancies. Therefore, it is not necessarily applicable to normal pregnancies.Thresholds for abnormal ABPM in pregnant women have been not defined; values defined for general population are not necessarily appropriate. However, a recently published study of an at-risk pregnant women in a southern Chinese population defined similar ABPM thresholds based in maternal and fetal outcomes [64].Therapeutics goals for ABPM have not been defined. Consequently, randomized clinical trials are necessary in order to define clarify this issue.

## 8. Conclusions

In conclusion, ABPM has a defined role in the evaluation of BP levels in pregnant women. We propose performing an ABPM before 20 weeks of gestation to appropriate classification of HDP, discarding white-coat hypertension and detecting masked chronic hypertension in pregnant women with office BP > 125/75 mmHg. Also, a second ABMP performed between 20–30 weeks of gestation allows for the detection of women with a high risk of development of PE, particularly early onset PE. Preliminary data and indirect evidence suggest that ABPM is also useful to evaluate hypertension control in pregnant women with treated chronic hypertension. Finally, in women who had PE, a third ABPM in the post-partum period could identify those with higher long-term cardiovascular risk related with masked hypertension (Table 2).

## Figures and Tables

**Figure 1 diagnostics-13-01457-f001:**
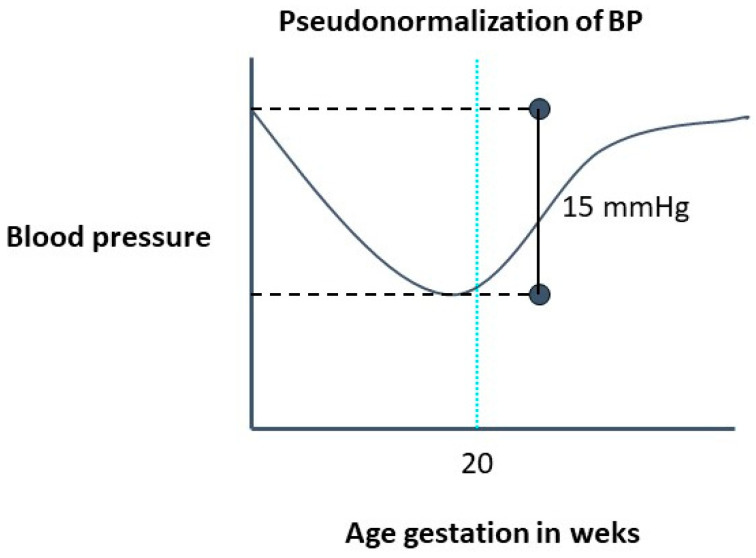
Evolution of blood pressure in a normal pregnancy.

**Figure 2 diagnostics-13-01457-f002:**
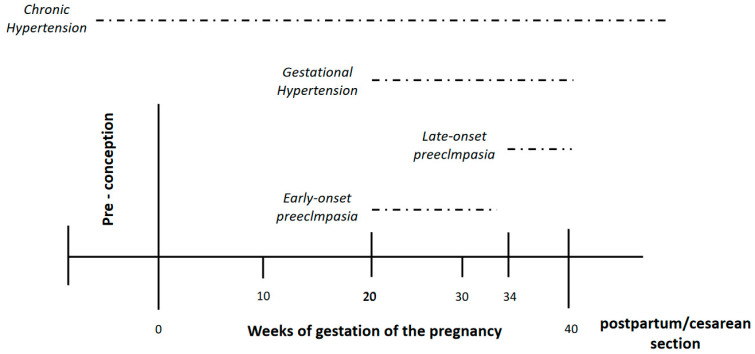
Hypertensive disorder pregnancy according to weeks of gestation.

**Table 1 diagnostics-13-01457-t001:** Absolute risk and odd-ratios for preeclampsia/eclampsia according to quartiles of systolic and diastolic office blood pressure (adapted Salazar et al. [16]).

	Office BP (mmHg)	Absolute Risk	Unadjusted Relative Risk		Adjusted Relative Risk *	
	Mean ± SD	(%)	OR	95% CI	OR	95% CI
SBP quartiles						
91.0–115.7 mmHg	108.7 ± 5.4	8.8	1		1	
116.0–124.7 mmHg	120.4 ± 2.8	13.4	1.60	0.63–4.08	1.35	0.52–3.51
125.0–132.7 mmHg	128.9 ± 2.2	19.6	2.52	1.04–6.14	2.17	0.87–5.43
133.0–165.0 mmHg	143.6 ± 8.1	32.3	4.94	2.12–11.51	3.99	1.66–9.56
		*p* < 0.001	*p* for trend = 0.001		*p* for trend = 0.004	
DBP quartiles						
48.7–69.3 mmHg	63.8 ± 4.5	6.5	1		1	
69.7–75.7 mmHg	72.7 ± 1.7	13.7	2.30	0.83–6.33	2.13	0.76–5.97
76.0–81.7 mmHg	78.9 ± 1.7	19.6	3.53	1.33–9.34	3.08	1.14–8.31
82.0–108.3 mmHg	88.8 ± 5.6	34.4	7.61	3.00–19.31	6.47	2.49–16.82
		*p* < 0.001	*p* for trend < 0.001		*p* for trend < 0.001	

* Adjusted for maternal age, gestational age at ambulatory blood pressure monitoring, multiparity, chronic kidney disease, history of preeclampsia, diabetes, chronic hypertension, antihypertensive treatment, low doses of acetylsalicylic acid, and calcium supplement. BP, blood pressure.

**Table 2 diagnostics-13-01457-t002:** Proposed indications for the use of ABPM in pregnant women.

Period of Gestation	Indications	Aims
First half	All women with office BP > 125/75 mmHg	To make an appropriate diagnose of chronic hypertension identifying white coat hypertension and masked hypertension.
Second half	Women with high-risk pregnancy	To identify women with high risk for preeclampsia.
Post-partum (3–6 months after the delivery)	Women who developed preeclampsia	To evaluate long-term cardiovascular prognosis identifying masked and nocturnal hypertension.

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
