# Peer review of "Ambulatory Blood Pressure Monitoring for Diagnosis and Management of Hypertension in Pregnant Women"

_diagnostics, 2023, doi:10.3390/diagnostics13081457_

Round 1

Reviewer 1 Report

The authors have tried to write an overview of current issues related to blood pressure monitoring in diagnosis and management of hypertension in pregnant women. They try to cover many aspects of the issue and review the literature, but they did not avoid errors that should be corrected before publishing this type of document.

Examples: 

Lines 34 - 35: I was unable to locate in the cited source [6] the information to reproduce Figure 1. 

Lines 99 - 100: The citation to [16] and Table 1 was presented as if Table 1 was part of the current article instead of [16]. 

Lines 149-152: Enigmatic information "Espeche recently Published" without any reference.

Lines 360-372: The conclusion is vague and I did not quite find the arguments for it in the document except for the line in Figure 1 indicating a value of 20, which is also of enigmatic origin.

Language errors:

Example Line 369: Grammar: "should be highlighted".

In my opinion, the article should be corrected. The attempt to show the various aspects of pregnancy hypertension should be presented in a more cause-and-effect form, from which clear conclusions should be drawn. In the current version of the article, I perceive a somewhat chaotic demonstration of several aspects divided into chapters, which is not comfortable for the reader. 

Author Response

The authors have attempted to write an overview of current issues related to blood pressure control in the diagnosis and management of hypertension in pregnant women. They try to cover many aspects of the subject and review the literature, but they do not avoid errors that must be corrected before publishing this type of document.

Examples:

Lines 34 - 35: I could not locate in the cited source [6] the information to reproduce Figure 1.

This bug was fixed. Thank you so much for the correction.

Lines 99 - 100: The citation of [16] and Table 1 was presented as if Table 1 was part of the current article instead of [16].

The paragraph was corrected. Thank you very much for your correction.

Lines 149-152: Enigmatic information "Espeche published recently" without any reference.

The reference was placed in the corresponding place. Thank you so much for the correction.

Lines 360-372: The conclusion is vague and I didn't find the arguments for it in the paper except for the line in Figure 1 indicating a value of 20, which is also of enigmatic origin.

The paragraph was changed with its respective reference. Figure 1 is on his own authority. Thank you so much for the correction.

Language errors:

Example line 369: Grammar: "should be highlighted."

It was corrected and also the writing was evaluated by a native English language translator. Thank you very much for the correction.

In my opinion, the article should be corrected. The attempt to show the various aspects of hypertension during pregnancy must be presented in a more cause and effect form, from which clear conclusions must be drawn. In the current version of the article, I perceive a somewhat chaotic demonstration of various aspects divided into chapters, which is not comfortable for the reader.

Unfortunately, the organization of the article cannot be modified, since it should be written in its entirety and we believe that with the changes suggested by you, the article now has greater clarity. Thanks for making the writing.

Reviewer 2 Report

I have read with interest this review regarding the utility of ambulatory blood pressure monitoring to diagnose and management of hypertension in pregnant women. Hypertension disorders of pregnancy are increasingly studied, and this article brings important data in this field. I consider that the article may be accepted for publication, after some minor revisions.

1. I consider it is important to introduce some tables or figures to resume the most important findings of this study. 

2. I suggest addressing the future scope and topics that are important and that could not be covered in the manuscript.

3. Please insert the limitations of this study.

4. The references need to respect the recommendations of the journal.  

Author Response

I have read with interest this review regarding the utility of ambulatory blood pressure monitoring to diagnose and management of hypertension in pregnant women. Hypertension disorders of pregnancy are increasingly studied, and this article brings important data in this field. I consider that the article may be accepted for publication, after some minor revisions.

  1. I consider it is important to introduce some tables or figures to resume the most important findings of this study.

Thanks for the suggestions, we added a table and a graph.

  1.  I suggest addressing the future scope and topics that are important and that could not be covered in the manuscript.

Thank you very much for the suggestion, we added this section.

  1. Please insert the limitations of this study.

Thank you very much, we added that suggestion.

  1. The references need to respect the recommendations of the journal.  

Thank you very much, the references were changed.

Round 2

Reviewer 1 Report

The authors took into account the suggestions and corrected the manuscript which is now much clearer and more readable. I believe that in its current form it is suitable for publication.

Author Response

Thank you very much for your suggestions.